# The genotype distribution, infection stage and drug resistance mutation profile of human immunodeficiency virus-1 among the infected blood donors from five Chinese blood centers, 2014–2017

Shan Liang[1,2©], Zhiyang Liu[1,2©], Shaoli Wang[1,2], Jing Liu[3], Ling Shi[4], Wei Mao[5], Cunxu Liu[6], Jianhua Wan[7], Lili Zhu[8], Mei Huang[9], Yu Liu[1,2], Jingxing Wang[1,2], Paul Ness[3], Hua Shan[10], Peibin Zeng[11]*, Miao He[1,2]*

1 Institute of Blood Transfusion, Chinese Academy of Medical Sciences, Chengdu, China, 2 Sichuan Blood Safety and Blood Substitute International Science and Technology Cooperation Base, Chengdu, China, 3 The Johns Hopkins Medical Institutions, Baltimore, Maryland, United States of America, 4 University of Massachusetts at Boston, Boston, Massachusetts, United States of America, 5 Chongqing Blood Center, Chongqing, China, 6 Guangxi Blood Center, Liuzhou, Guangxi, China, 7 Urumqi Blood Center, Urumqi, Xinjiang, China, 8 Luoyang Blood Center, Luoyang, Henan, China, 9 Mianyang Blood Center, Mianyang, Sichuan, China, 10 Stanford University, Stanford, California, United States of America, 11 West China School of Public Health and West China Fourth Hospital, Sichuan University, Chengdu, Sichuan, China

© These authors contributed equally to this work.
* zengpeibin@live.cn (PZ); hemiao@ibt.pumc.edu.cn (MH)

**Data Availability Statement:** The HIV-1 sequences in the study can be retrieved from GenBank with

## Abstract

Human immunodeficiency virus-1 (HIV-1) exhibits high diversity and complexity in China, challenging the disease surveillance and antiretroviral therapy. Between July 1, 2014 and January 30, 2017, we investigated the profiles of HIV-1 infection stages, genotype distribution and drug resistance mutations (DRMs) using plasma samples from HIV Western blot (WB) confirmed blood donors from five Chinese blood centers (Chongqing, Guangxi, Luoyang, Mianyang, and Urumqi). HIV *pol* regions consisted of whole protease and partial reverse transcriptase were genotyped and analyzed for DRMs. Lag-Avidity testing was performed to identify the infection stages. Of the 356 HIV-1 WB positive samples tested by Lag-avidity assay, 19.1% (68/356) were recent infections. Genotyping on 356 amplified sequences presented the subtype distributions as following: CRF07_BC (65.7%), CRF08_BC (7.3%), CRF01_AE (19.1%), B (4.2%), CRF55_01B (3.1%), CRF59_01B (0.3%) and CRF68_01B (0.3%). No significant difference in genotype distribution was observed between recent and long-term infections. 48 DRMs were identified from 43 samples, indicating a drug resistance prevalence of 12.1% (43/356), which include seven protease inhibitors (PIs) accessory DRMs (Q58E, L23I and I84M), two PIs major DRMs (M46I, M46L), seven nucleoside RT inhibitors DRMs (D67N, K70Q, K219R and M184L), and 32 non-nucleoside RT inhibitors DRMs (K103N, V179E, K238N, V179D, E138G, G190E, A98G, Y188D and E138A). In addition, we had also identified CRFs from the 01B subtype including CRF55_01B (3.1%), CRF59_01B (0.3%) and CRF68_01B (0.3%). As an important part of the continuous monitoring of HIV-1 circulating strains among blood donors, our

accession numbers from MW232549 to MW232904.

**Funding:** The study was supported by the Recipient Epidemiology and Donor Evaluation Study-III (REDS-III) International Component (China) (HHSN268201100008I) and the CAMS Innovation Fund for Medical Sciences (CIFMS) (CAMS-2016-I2M-3-025) to all authors, and the Basic research plan of Guizhou science and Technology Department to PZ (Qian ke he ji chu [2018]1095).

**Competing interests:** The authors have declared that no competing interests exist.

findings were expected to contribute to the comprehensive AIDS control and development of proper diagnostics for HIV-1 in China.

## Introduction

The spread of HIV infection continues to pose a significant public health threat in China as well as globally [1,2]. According to the latest China Statistical Yearbook, nationally reported AIDS cases in 2018 were 64170, the incidence rate was 4.6199/100,000, the number of deaths was 18780, and the death rate was 1.3459/100,000 [3]. Studies have shown that the HIV-1 epidemic has spread from high-risk groups to the general population including Chinese blood donors [4]. The predominant genotypes of HIV-1 in the general population in China include circulating recombinant form (CRF) 07_BC, CRF08_BC, CRF01_AE and subtype B [5–7]. For a long time, the infected population and HIV gene subtypes in China have been constantly changing, so the epidemiological study on AIDS is helpful to better monitor the epidemic of AIDS in China.

Although the risk of transfusion transfusion-transmitted HIV infection in China has been significantly reduced in the past decades [8], the incidence and prevalence of HIV among Chinese blood donors are still persistent [9]. Studying HIV genotypic characteristics and profiles of drug resistant mutations (DRMs) in blood donors is an important part of an ongoing HIV molecular surveillance program and critical for developing appropriate testing and [7] treatment programs targeting the current and dominating strains. It has been described that the characteristics of HIV-1 genotype distribution evolved and diversified between different regions and populations [10]. Previous study during 2012 to 2014 from the NHLBL Recipient Epidemiology and Donor Evaluation Study-III (REDS-III) program reported the genotype distribution of HIV-1 infected donors from five blood centers (Chongqing, Guangxi, Luoyang, Mianyang, and Urumqi): CRF07_BC (65.7%), CRF08_BC (7.3%), CRF01_AE (19.1%), B (4.2%), and 01B (3.7%) [7].

As an important part of continued surveillance in REDS-III program, the current study updated the findings of HIV infection stages, genotype distribution and characteristics of DRMs from July 1, 2014 to January 30, 2017 among HIV infected donors from five Chinese blood centers. The findings may lead to better understanding on the HIV-1 molecular characteristics and help on the improvement of HIV-1 diagnostic and blood screening in China.

## Materials and methods

This study was approved by institutional review board (IRB) of Johns Hopkins Medicine, NA_00080591/ CR00012868 S1 Text and ethical review committee of Chinese Academy of Medical Sciences/Pekin Union Medical College, X101222002 S1 Fig (In written form). In order to protect the privacy of blood donors, the sample was anonymous before it was obtained, X101222002 S1 Fig (In written form). We have obtained informed consent from the donor, JL/LYXZ-C-11-062 S2 Fig (in writing form).

### Study samples

Approximately 350,000 donations per year were collected from these five blood centers, accounting for 3% of the total national blood donations. From July 2014 to January 2017, routine parallel enzyme-linked immunosorbent assay (ELISA) screening was performed for each donation using previously described two of the six assays at each blood center [11]. ELISA

screening reactive samples were shipped via cold chain to Institute of Blood Transfusion (IBT) for confirmatory testing by Western Blot (WB)(MP Diagnostics HIV BLOT 2.2, MP Biomedicals Asia Pacific Pte Ltd, Singapore). The consents from participated donors were obtained before testing.

### Extraction, amplification, sequencing of HIV-1 RNA

RNA extraction from plasma was performed using a viral RNA isolation kit (MagMAX, Ambion, Inc., Austin, TX, USA). Then HIV *pol* region which includes partial reverse transcriptase gene and the whole protease gene was amplified by reverse transcription nested polymerase chain reaction (RT nest PCR) following previously reported methods [12,13]. Amplified samples were sent to a commercial sequencing company for sequence determination (Tsing Ke Biotechnology Co. Ltd, Chengdu, China).

### Genotype, phylogenetic analysis and DRM analysis

We used MEGA 7.0 (http://www.megasoftware.net) to align the sequences and build the phylogenetic tree using the neighbor-joining method. The HIV reference sequence were obtained from the Los Alamos database (http://www.hiv.lanl.gov) as the following (Genbank Accession Number): CRF07_BC: AF503396, JF906666, HQ215583, AF286226, HG421616, JQ901027, JQ901094, EF122515; CRF01_AE: GU564223, JX112798, AY008714, U51189, JF906597, JQ028206; CRF08_BC: AF286229, AY008717; Subtype B: K03455, U71182, AY173951, AY33195; CRF55_01B: JX574662, JX574663, JX960635, JX574661, KF857447; CRF69_01B: AB845344, AB845345; CRF62_BC: KC870035, KC870034, KC870037; CRF68_01B: KC183783; CRF59_01B: JX960635, KC46219, JX574661; The HIV DRMS were determined and analyzed by submitting the sequences to the Stanford HIVdb Program Genotypic Resistance Interpretation Algorithm (http://hivdb.stanford.edu, HIVdb version 8.7, last updated: 2018-10-19).

### HIV infection stage determination

The Lag-Avidity EIA kit (HIV-1LAg-Avidity EIA, KINGHAWK, China) was used to determine HIV infection stages [14]. The test is an accurate serological marker that is widely used in the study of infection stage and can easily and quickly estimate the incidence of AIDS, This method has been used for HIV infection staging in blood donors and defines recent infection as infection within 130 days [7].

### Statistical analysis

Donor demographic information was extracted from the REDS-III blood donation database. Statistical analysis was performed using the SPSS 17.0. Chi-square tests were performed to compare the differences in drug resistance at the infection stage. $P<0.05$ was defined as statistically significant.

### Gene accession numbers

The HIV-1 sequences in the study can be retrieved from GenBank with accession numbers from MW232549 to MW232904 (S2 Text).

## Results

A total of 356 HIV WB-confirmed donation samples were collected from five REDS-III blood centers including Chongqing (n = 207), Urumqi (n = 54), Mianyang (n = 23), Luoyang

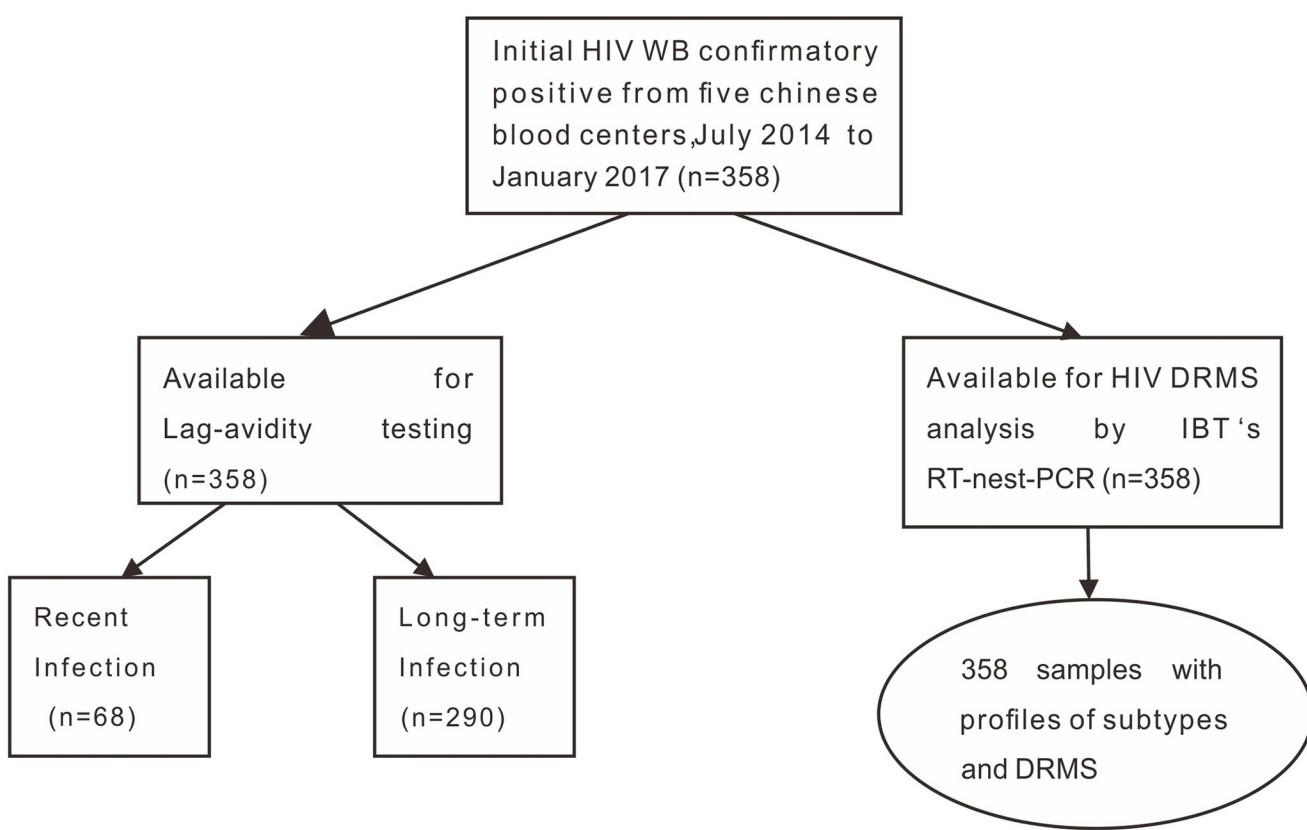

**Fig 1. Summary of HIV testing and DRMS analysis among blood donors from five Chinese blood centers.** July 2014 to January 2017.

(n = 20), Guangxi (n = 52). 356 samples were tested by Lag-avidity assay for HIV infection stage assignment. The testing algorithm is summarized in Fig 1.

## HIV Genotype/CRF distributions

HIV partial *pol* sequences from 356 HIV confirmed positive samples were successfully amplified, sequenced and analyzed for HIV subtype/CRF. HIV subtype information of the 356 donors from five blood centers, including: Chongqing (n = 207), Urumqi (n = 54), Guangxi (n = 52), Luoyang (n = 20) and Mianyang (n = 23), is displayed in Table 1.

Distribution of all HIV-1 subtypes by phylogenetic analysis is displayed in Fig 2. CRF07_BC (65.7%) and CRF01_AE (19.1%) were two major CRFs. The rest were CRF08_BC

**Table 1. Subtypes of 356 infected donors whose HIV genotypes and DRMs were successfully analyzed.**

| | | Chongqing N = 207 | Urumqi N = 54 | Guangxi N = 52 | Luoyang N = 20 | Mianyang N = 23 | Total N = 356 |
|---|---|---|---|---|---|---|---|
| HIV subtype/CRF | CRF07_BC | 156 | 41 | 11 | 10 | 16 | 234 |
| | CRF01_AE | 19 | 5 | 36 | 3 | 5 | 68 |
| | CRF08_BC | 21 | 1 | 4 | 0 | 0 | 26 |
| | CRF55_01B | 3 | 4 | 0 | 3 | 1 | 11 |
| | B | 7 | 2 | 1 | 4 | 1 | 15 |
| | CRF68_01B | 1 | 0 | 0 | 0 | 0 | 1 |
| | CRF59_01B | 0 | 1 | 0 | 0 | 0 | 1 |

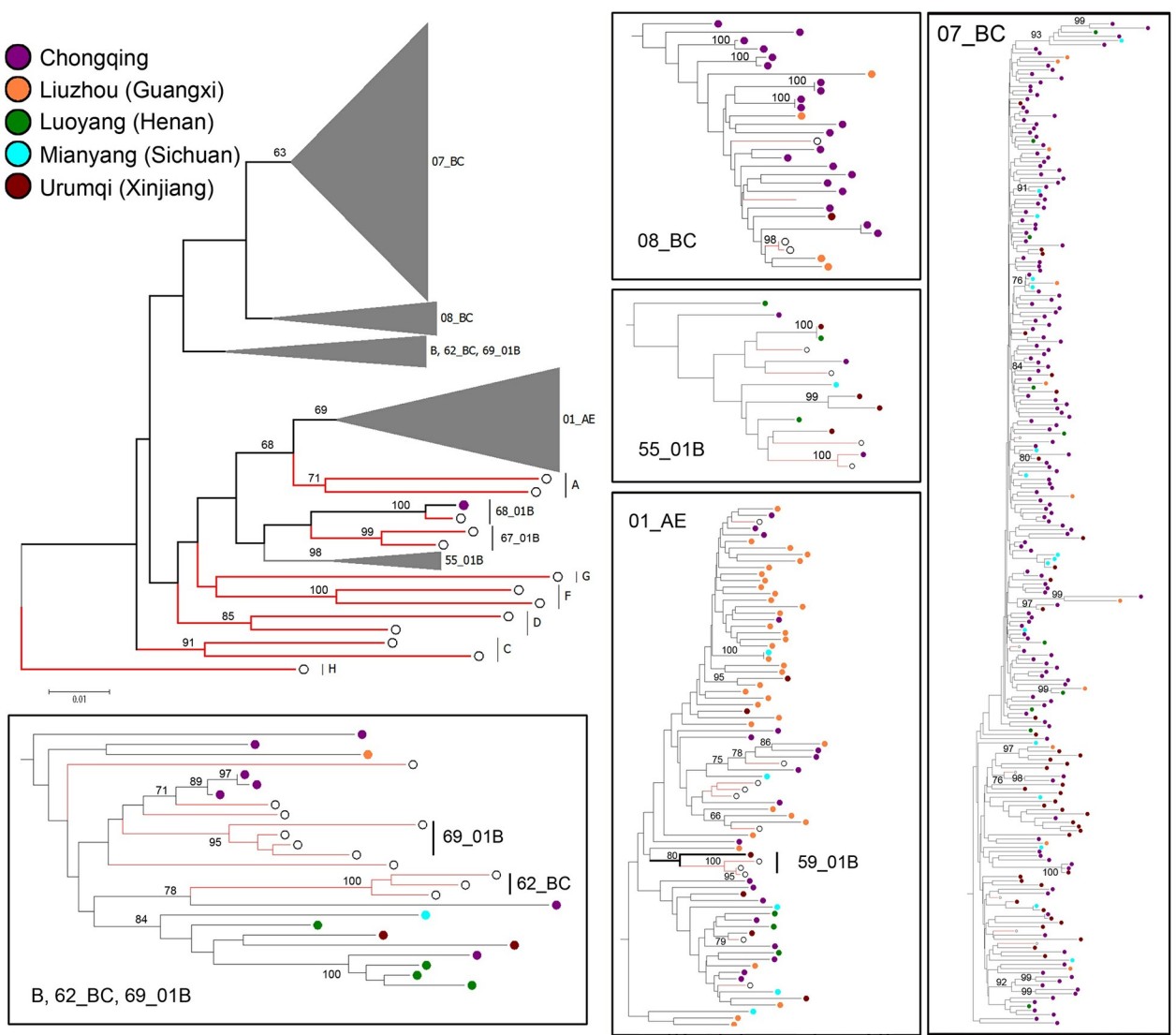

**Fig 2. Distribution of all HIV-1 subtypes by phylogenetic analysis (Bootstrap value cant difference of HIV-1 subty.**

(7.3%), B (4.2%), 55_01B (3.1%), 59_01B (0.3%) and 68_01B (0.3%). The HIV-1 subtype distribution in Chongqing was as follows: CRF07_BC = 156 (75.4%), CRF08_BC = 21 (10.1%), CRF01_ AE = 19 (9.2%), B = 7 (3.4%), CRF55_01B = 3 (1.4%) and CRF 68_01B = 1 (0.5%). The dominant subtypes in Urumqi and Guangxi were CRF07_BC (76.0%, 41/54) and CRF01_AE (69.2%, 36/52), respectively. Among these subtypes, CRF55_01B, CRF59_01B and CRF68_01B are the new recombination types discovered in this study. No significant difference of HIV-1 subtype's distribution was observed between recent and long-term infections.

## HIV infection stage determination

Of the 356 HIV confirmatory positive samples tested by Lag-avidity assay, 68 (19.1%) were identified as recent infection and 288 (80.9%) as long-term infection. The recent infection rate of each genotype in Chongqing (Table 2): CRF07_BC (17.3%), CRF01_AE (21.1%), CRF08_BC (33.3%), B (57.1%), CRF55_01B (33.3%), CRF68_01B (0). Recent infection rate in

**Table 2. Infection stages of 356 HIV infected donors from five Chinese blood centers (N, %) [a].**

| Blood centers | | Subtype | | | | | | |
|---|---|---|---|---|---|---|---|---|
| | | CRF07_BC | CRF01_AE | CRF08_BC | B | 55_01B | 59_01B | 68_01B |
| Chongqing | Recent | 27 (17.3%) | 4 (21.1%) | 7 (33.3%) | 4 (57.1%) | 1 (33.3%) | 0 (0) | 0 (0) |
| | Long-term | 129 (82.7%) | 15 (78.9%) | 14 (66.7%) | 3 (42.9%) | 2 (66.7%) | 0 (0) | 1 (100%) |
| Urumqi | Recent | 5 (12.2%) | 2 (40%) | 0 (0) | 0 (0) | 0 (0) | 0 (0) | 0 (0) |
| | Long-term | 36 (87.8%) | 3 (60%) | 1 (100%) | 2 (100%) | 4 (100%) | 1 (100%) | 0 (0) |
| Guangxi | Recent | 0 (0) | 6 (16.7%) | 1 (25%) | 0 (0) | 0 (0) | 0 (0) | 0 (0) |
| | Long-term | 11 (100%) | 30 (83.3%) | 3 (75%) | 1 (100%) | 0 (0) | 0 (0) | 0 (0) |
| Luoyang | Recent | 4 (40%) | 1 (33.3%) | 0 (0) | 0 (0) | 0 (0) | 0 (0) | 0 (0) |
| | Long-term | 6 (60%) | 2 (66.7%) | 0 (0) | 4 (100%) | 3 (100%) | 0 (0) | 0 (0) |
| Mianyang | Recent | 4 (25%) | 1 (20%) | 0 (0) | 0 (0) | 1 (100%) | 0 (0) | 0 (0) |
| | Long-term | 12 (75%) | 4 (80%) | 0 (0) | 1 (100%) | 0 (0) | 0 (0) | 0 (0) |
| In total | Recent | 40 (17.1%) | 14 (20.6%) | 8 (30.8%) | 4 (26.7%) | 2 (18.2%) | 0 (0) | 0 (0) |
| | Long-term | 194 (82.9%) | 54 (79.4%) | 18 (69.2%) | 11 (73.3%) | 9 (81.8%) | 1 (100%) | 1 (100%) |

[a]Data are reported as number and percent of total within each blood center.

Urumqi were CRF07_BC (12.2%), CRF01_AE (33.3%). Recent infection rates in Guangxi were mainly concentrated in two genotypes: CRF01_AE (16.7%) and CRF08_BC (25%). The recent infection rate in Luoyang were CRF07_BC (40%), CRF01_AE (33.3%). Mianyang recent infection rate were CRF07_BC (25%), CRF01_AE (20%), CRF55_01B (100%). There is no significant difference in the recent infection rate among subtypes in different regions.

## DRMs profiles

48 DRMs were identified from 43 samples including seven protease inhibitors (PIs) accessory DRMs, two PIs major DRMs, seven nucleoside RT inhibitors DRMs, and 32 non-nucleoside RT inhibitors DRMs. 43 samples had DRMs, indicating a drug resistance prevalence of 12.1% (43/356). We analyzed the following specific DRMs: 1) 66.7% (32/48) DRMs were on nonnucleoside reverse transcriptase inhibitors (NNRTIs) as: V179E (n = 12), E138A (n = 4), Y188D (n = 1), K103N (n = 2), A98G (n = 1), K238N (n = 2), V179D (n = 8), E138G (n = 1), G190E (n = 1). 2) Seven nucleotide reverse transcriptase inhibitors (NRTIs) were: D67N (n = 4), M184L (n = 1), K70Q (n = 1), K219R (n = 1). 3) Nine protease inhibitor (PIs) DRMs were observed including, seven PIs accessory DRMs: Q58E (n = 5), L23I (n = 1), I84M (n = 1) and two PIs major DRM: M46L (n = 2). The rates of HIV-1 DR varied by blood centers with the following distribution: Chongqing (12.6%, 26/207), Guangxi (3.8%, 2/52), Urumqi (16.7%, 9/54), and Luoyang (15%, 3/20), and Mianyang (13.0%, 3/23). Summary of the DRMs is displayed in S1 Table.

## Discussion

Twenty years ago, the first nationwide survey reported the predominant HIV genotype in China was subtype B'/B, subtype C, and CRF01_AE [15]. In recent years, the prevalence of HIV-1 genotypes among the general population from high to low was CRF01_AE, CRF07_BC, subtype B'/B, CRF08_BC, subtype C, followed by CRFs [16]. CRFs had become the predominant genotype. Blood donors in our research were from regions with the highest HIV prevalence in western and central China [17,18]. In our previous study, the proportion of each genotype was: CRF07_BC = 234 (65.7%), CRF08_BC = 26 (7.3%), CRF01_AE = 68 (19.1%),

B = 15 (4.2%), and 01B = 13 (3.7%). The most prevalent subtype of HIV-1 in our present study was CRF07_BC (65.7%) and CRF01_AE (19.1%), CRF07_BC was higher than that in the previous study (65.7% vs. 61.5) [7]; CRF01_AE was mainly concentrated in Guangxi and lightly lower (19.1% vs.20%). The main subtypes of Urumqi and Mianyang are consistent with the previous research of REDS-III: CRF07_BC and CRF01_AE. The main subtype of the western region (Chongqing, Urumqi, Mianyang) is CRF07_BC, which is consistent with the previous part. The main genotype in Guangxi in the chinese western region is also CRF01_AE, which is due to the fact that Guangxi is adjacent to Vietnam and Yunnan, with more injection drug users and men who have sex men (MSM) [19]. According to the phylogenetic analysis, some new epidemic CRFs (marked as 01B in previous study) in this project including: CRF55_01B, CRF59_01B and CRF68_01B were found in the Chinese blood donors. These CRFs were proved highly prevalent in MSM:

1. CRF55_01B, one of the first CRFs circulating mostly among MSM in China [20], was firstly reported by Han [21], composed of CRF01_AE and subtype B with four recombination breakpoints in the *pol* gene, and then a large-scale followed survey of CRF55_01B among MSM [22] showed its origin was in 2000 and had spread throughout the most provinces.

2. CRF68_01B was first found in the MSM population of Anhui province in China [23], and was composed of CRF01_AE and subtype B. A large number of studies found that it was mainly concentrated in the southeast, such as Jiangsu [24], Shanghai [25], Zhejiang [26] and other places, especially in economically developed regions. At present, CRF68_01B is mainly spread across regions by MSM population.

Finding some cases of these three CRFs in our research suggests that the HIV epidemic is spreading from high-risk groups (e.g., MSM) to the general population, including blood donors. In recent years, MSM has become the major mode of HIV transmission [17,27], These new CRFS have been found in other provinces and cities in China. Through the study, it is found that most MSM are already married [28], thus causing the transmission of specific HIV-1 subtypes into the general population, and always been proved as a high-risk population of HIV transmission in blood donations domestically and internationally [4,29]. To minimize the risk of HIV transmission from MSM blood donations, we should focus more on blood donor questionnaire design about MSM behavior, even it is a culturally sensitive topic in China.

The rate of HIV-1 DR rate was 12.1% (43/356) is similar to that reported in the United States (13.0%) [30], Brazil (16.3%) [31] and some European countries such as Spain (14.0%) [32], but higher than Asia (4.6%) [33], The national (4.5%) [34], Chengdu (1.3%) [12], Guangxi (3.2%) [35] and five Blood center in 2007–2010 (4.4%) (Liuzhou, Kunming, Urumqi, Mianyang, Luoyang) [36], the difference may be caused by the continuous application of anti-retroviral therapy (ART), so increasing the resistance of HIV-1. In this study, the HIV drug resistance rate in Chongqing, Urumqi and Mianyang was roughly the same as in previous studies. The DRMs prevalence rate reported here (12.1%, 43/356) was similar to our previous study (13.2%, 27/205) [7]. In Urumqi, the HIV-1 DR rate was 16.7% (9/54) slightly lower than previous study (24.1%, 7/29). The HIV drug resistance rate in Liuzhou increased from 0% to 3.8%, which was corresponding to the main genotype CRF01_AE in Liuzhou. Due to the special geographical location of Guangxi, intravenous drug users and MSM increased. In HIV drug resistance mutation in the study, 66.7% of the mutations for non-nucleoside reverse transcriptase inhibitors (NNRTI) DRM, These mutations will lead to clinical resistance to drugs such as nevirapine (NVP) and efavirenz (EFV). In the study of the same five central blood stations from 2012 to 2014 [7], it was found that V179E was still the main mutation in the current

study in NNRTI DRM. Therefore, continuous detection of genotype resistance mutations can help prevent and reduce drug resistance caused by genotype mutations, so as to better control the transmission and treatment of HIV drug resistance.

Besides, the results of the current study indicate that there is no significant difference in drug resistance between recent and long-term infections (P>0.05) (Table 3).

In our study, most of the samples had DR due to DRMs on NNRTIs (32/48). According to the Stanford HIVdb program, drug-specific NNRTIs mutations are associated with the selection of patients receiving nevirapine (NVP) and efeviren (EFV). These two drugs are often used as first-line NNRTIs in antiretroviral therapy in China [37]. As for PIs, they are not included as the first line ART drugs in China. In recent years, because highly active anti-retroviral treatment (HAART) is widely used in clinical practice, HIV has developed resistance to many drugs, which also brings great challenges to clinical treatment. The study of drug resistance mutation of HIV subtypes is crucial to the monitoring of drug resistance.

Our previous HIV-1 molecular epidemiology study of infected donors at the same five blood centers between 2012 and 2014 showed an average recent infection rate of 23.6% (61/ 259) [7]. The present study yielded a recent infections' rate 19.1%. Previous studies have shown that the recent HIV infection rate in China is increasing year by year and is closely related to the MSM population and young people under the age of 30 [38]. This is because many MSM population are more aware of being actively tested for HIV than the general population. According to Table 1, the recent rates of CRF07_BC, CRF01_AE, B, and CRF55_01B are roughly the same, while the recent rates of CRF08_BC are slightly higher. Moreover, the genotypes of HIV in China are still dominated by CRF07_BC, CRF01_AE and CRF08_BC [39–41]. This is similar to the distribution of subtypes of newly diagnosed HIV infection in China in 2015 [42]. Studies have shown that these genotypes are the main subtypes of current infection in China. It was first discovered among injecting drug users in Yunnan province [43]. In our study, the subtypes with the most complex recent rates were detected in the blood donor population in Chongqing area. CRF07_BC, CRF08_BC and CRF01_AE were the most common subtypes in recent infections. CRF55_01B and CRF59_01B were the emerging subtypes in this study. The results are in line with the characteristics of Chongqing's growing MSM population. The CRF08_BC genotype in Urumqi was detected in a recent infection. Mianyang's CRF55_01B is the newly detected genotype and was found in a recent infection. Chongqing is one of the regions with high incidence of AIDS in China. Due to the economic development, the floating population is gradually increasing, resulting in the increase of a large number of MSM population in Chongqing. The overall infection rate of MSM population from 2013 to 2017 was 20.5% [44], much higher than that of MSM population in China (6.3%) [45]. In addition, the incidence of MSM in outlander is higher than that of local population [26]. In one study, the recent HIV infection rate of MSM in Chongqing (21.2%) was much higher than that of men in the same period (9.1%), suggesting that the recent infection rate reflects the changes of MSM population in society [45]. It can be inferred that most of the recent infections of the genotypes in our results are from MSM, which is also consistent with the prevalence of these genotypes. In our results, the newly discovered genotypes are all from MSM [40,41], indicating that the genotypes of MSM population infected with AIDS are

**Table 3. Differences in drug resistance between recent and long-term infections.**

|  | Recent | Long-term | Total | P value |
|---|---|---|---|---|
| Sensitivity | 62 | 250 | 312 | 0.355 |
| Resistance | 6 | 37 | 43 |  |
| Total | 68 | 288 | 355 |  |

becoming more and more complex, which brings great challenges to the treatment and prevention of AIDS.

In recent years, surveys of blood donors have found that young people (less than 25 years old) are the main group of AIDS patients, accounting for as much as 54.4% [46]. The main reason for the increasing rate of young people infected with AIDS is that with the development of socio-economic culture, young people are becoming more open-minded, like to pursue stimulus, and lack of rational judgment of sex [47,48]. MSM, as well as women in sexual business, mainly young people, usually have a personality trait: sexual sensation seeking (SSS) [49]. And research shows that the highest recent infection rate in China is young people under 25 years old [44]. At the same time, there has been a high increase in AIDS among the elderly, mainly in heterosexual transmission [50]. China has the largest group of the elder among worldwide [51]. The proportion of reported HIV infections among the elderly has been increasing in China in recent years [52] which has been postulated as related to increased commercial sexual activities after retirement [53–56]. These phenomena require us to pay more attention to special groups.

Research shows that many countries have enacted the law of delaying blood donation to assess MSM, so as to determine the time limit for delaying blood donation [57–59]. Since 1998, China has enacted a new blood donation law that prohibits MSM people from donating blood, so it is impossible to predict how the delayed policy will work in China.

In addition to the MSM high-risk population, our research team has previously surveyed the risk factors of HIV infection among blood donors from seven blood centers. Studies have shown that men (87.2%) are much larger than women (12.8%), The proportion of blood donors with an education below secondary school is nearly half (41.2%), migrant workers and other unspecified occupation ratio is higher than control group, in the past six months the intramuscular (IM) or intravenous (IV) associated with HIV positive, and has been a diagnosis of sexually transmitted diseases such as syphilis or gonorrhea is associated with HIV positive [60]. Due to time, our study was not associated with these risk factors, but this result can also serve as a reference for this study.

Continued monitoring of molecular epidemiology characteristics of HIV infection in blood donors is an important component of a comprehensive HIV surveillance program, especially the HIV prevalence of specific populations that have been identified through research. Findings on the trend of these characteristics help to provide guidance on developing update policies and prevention measures to safeguard blood safety.

## Supporting information

**S1 Fig. IRB from CAMS.**
(TIF)

**S2 Fig. Blood donor health information request form.**
(TIF)

**S1 Table. HIV-1 genotypic drug resistance mutations.**
(DOC)

**S1 Text. IRB from JHM.**
(PDF)

**S2 Text. 356 sequences from Chinese HIV-1 infected donors.**
(TXT)

**S1 Excel. China REDSIII HIV testing summary.**
(XLSX)

**S1 File. The parameters for the phylogenetic analysis.**
(MAS)

## Author Contributions

**Conceptualization:** Shan Liang, Ling Shi, Peibin Zeng, Miao He.

**Data curation:** Zhiyang Liu, Shaoli Wang, Wei Mao, Cunxu Liu, Jianhua Wan, Lili Zhu, Mei Huang, Hua Shan, Peibin Zeng.

**Formal analysis:** Shan Liang, Shaoli Wang, Ling Shi, Yu Liu, Hua Shan.

**Funding acquisition:** Jing Liu, Ling Shi, Paul Ness, Hua Shan.

**Investigation:** Shan Liang, Zhiyang Liu, Jing Liu, Yu Liu, Jingxing Wang.

**Methodology:** Zhiyang Liu, Shaoli Wang, Ling Shi, Yu Liu, Hua Shan, Peibin Zeng, Miao He.

**Project administration:** Shan Liang, Ling Shi, Yu Liu, Jingxing Wang, Paul Ness, Miao He.

**Resources:** Zhiyang Liu, Jing Liu, Wei Mao, Cunxu Liu, Jianhua Wan, Lili Zhu, Mei Huang, Miao He.

**Software:** Zhiyang Liu, Shaoli Wang, Peibin Zeng.

**Supervision:** Shaoli Wang, Jing Liu, Jingxing Wang, Peibin Zeng.

**Validation:** Jingxing Wang, Paul Ness, Hua Shan.

**Visualization:** Jing Liu, Jingxing Wang, Peibin Zeng.

**Writing – original draft:** Shan Liang, Zhiyang Liu.

**Writing – review & editing:** Shan Liang.

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
