## [Decision Letter · Decision Letter 0]

27 Aug 2020

PONE-D-20-23288

The Genotype Distribution, Infection Stage and Drug Resistance Mutation Profile of Human Immunodeficiency Virus-1 among the Infected Blood Donors from Five Chinese Blood Centers, 2014-2017

PLOS ONE

Dear Dr. He,

Thank you for submitting your manuscript to PLOS ONE. After careful consideration, we feel that it has merit but does not fully meet PLOS ONE’s publication criteria as it currently stands. Therefore, we invite you to submit a revised version of the manuscript that addresses the points raised during the review process.

We look forward to receiving your revised manuscript.

Kind regards,

Chiyu Zhang, Ph.D.

Academic Editor

PLOS ONE

Journal Requirements:

'Human Subject Research

name：Chinese Academy of Medical sciences/Peking Union Medical College ethical reriew committee

the approval number:X101222002

form:written'

a. Please amend your current ethics statement to confirm that your named institutional review board or ethics committee specifically approved this study.

3. Please provide additional details regarding participant consent.

In the ethics statement in the Methods and online submission information, please ensure that you have specified what type you obtained (for instance, written or verbal, and if verbal, how it was documented and witnessed).

If your study included minors, state whether you obtained consent from parents or guardians.

If the need for consent was waived by the ethics committee, please include this information.

4. Please amend the manuscript submission data (via Edit Submission) to include authors Liu Zhiyang, Wang Shaoli, Liu Jing, Shi Ling, Mao Wei, Liu Cunxu, Wan Jianhua, Zhu Lili, Huang Mei, Liu Yu, Wang Jingxing, Paul Ness, Shan Hua and Zeng Peibin.

5. One of the noted authors is a group or consortium; NHLBI Recipient Epidemiology and Donor Evaluation Study-III program.

In addition to naming the author group, please list the individual authors and affiliations within this group in the acknowledgments section of your manuscript.

Please also indicate clearly a lead author for this group along with a contact email address.

6. Please include a separate caption for each figure in your manuscript.

Reviewers' comments:

Reviewer's Responses to Questions

**Comments to the Author**

1. Is the manuscript technically sound, and do the data support the conclusions?

Reviewer #1: Yes

Reviewer #2: Partly

2. Has the statistical analysis been performed appropriately and rigorously? 

Reviewer #1: Yes

Reviewer #2: Yes

3. Have the authors made all data underlying the findings in their manuscript fully available?

Reviewer #1: No

Reviewer #2: Yes

4. Is the manuscript presented in an intelligible fashion and written in standard English?

Reviewer #1: No

Reviewer #2: Yes

5. Review Comments to the Author

Reviewer #1: In this paper, the authors investigated the genotype distribution and drug resistance mutation profile of HIV-1 among HIV-1 positive patients in 5 different Chinese blood centers during July 1, 2014 - January 30, 2017. They obtained HIV-1 Pol sequences, subtyped them and analyzed the DRMs. These findings will be helpful for the prevention and control of HIV/AIDS in China.

Comments:

1)Materials and methods: The parameters for the phylogenetic analysis should be provided. The Genbank Accession Numbers of all obtained sequences should be provided. .

2)The authors mentioned that “Statistical analysis was performed using the chi-square test (or the Fisher's exact test) ” When the Fisher's exact test were used? In addition, “double-tailed test” should be “two-tailed test”..

3)Page 10, Line 184-189: The authors should carefully check the classification of DRM. for example, M184L belongs to NRTI, rather than NNRTI. “184M” should be written in a complete form.

4)Page 12, Lines 239: Reference 26 did not support the statement that “The study also points out that these new CRFS are all from MSM and spread across regions through MSM groups”.

5)Page 12, Lines 246: How the DR prevalence rate ere (12.8%, 20/156) to be calculated?

6)Page 12, Lines 249: The national HIV-1 drug resistance prevalence should be updated. Please see and cite the most recent Meta-analysis paper in EClinicalMedicine. 2020 Jan 5;18:100238.

7)Page 13, Lines 264: The statement that “the V179E may be transmitted” should be supported by phylogenetic analysis.

8)The detailed demographic characteristics (age, gender, education, occupation and so on), subtypes, and DRMs of recruited patients should be provided.

9)S1 Table: Wrong classification of DRMs and incomplete drug sensitivity of patients.

10)Language can be further improved.

Reviewer #2: Comments to the Author:

This paper reports characteristics of HIV-1 subtype distribution, infection stages and drug resistance among blood donors from five Chinese blood centers (Chongqing, Guangxi, Luoyang, Mianyang and Urumqi) in 2014-2017. However, the HIV-1 subtype data shown in Table 1 and Figure 2 are not consistent. So the results of the article seem unconvincing. A re-analysis of HIV-1 subtype data are suggested. Thus, my suggestion is the manuscript needs major revisions before it can be published. The following comments may be helpful for the revision of this paper.

Major comments:

1. The HIV-1 subtype data shown in Table 1 and Figure 2 are not consistent. In Table 1 and Figure 2, the total number of Subtypes B, CRF69_01B and CRF62_BC were 15 and 17, respectively. Sequence CRF69_01B from Chongqing unmarked the color of node. The caption of CRF62_BC only including the reference sequences CRF62_BC, not the CRF62_BC sequence from Chongqing. In the subtree of CRF08_BC, CRF08_BC sequence from Urumqi unmarked the color of node. In the subtree of CRF55_01B, a total of 12 CRF55_01B sequences, however, only 11 CRF55_01B sequences showed in Table 1. In the subtree of CRF01_AE, one sequence belongs to CRF59_01B, however, no CRF59_01B appeared in the manuscript. In Table 1 and Figure 2, the number of CRF07_BC distribution in Mianyang and Guangxi were 16, 13 and 17, 11, respectively. There are two CRF_07 BC reference sequences not marked with node symbols. Please check the data carefully.

2. In recent years, HIV-1 CRF55_01B has increased from the MSM population in China. In the present study, CRF55_01B was identified among blood donors in Chongqing (n=3), Urumqi (n=4), Luoyang (n=3) and Mianyang (n=1) (Table 1). I wonder there is the transmission link between the blood donors and MSMs, or link to CRF55_01B sequences from these regions and other region in China. So, I suggest authors try to perform a Bayesian analysis to infer the original time and transmission dynamics of the CRF55_01B of these blood donors, and elucidate their migration from these different regions in China or transmission between these two populations (blood donors and MSMs).

Minor comments:

1. Line 49: As the authors mentioned, “two PIs major DRMs (M46I)”, however, the PIs major DRMs shown in Table S1 is M46I/L. The authors should check this data.

2. Line 50, Line 181: “thirty-four” should be “34”.

3. Line52-53:CRF62_BC is identified from the 01B subtype? If so, the subtype information should be checked.

4. Line 66:“sub genotypes”should be “subtypes”or “genotypes”.

5. Line 84-89: The font of the paragraph needs to be reformatted, so that the paragraph format is consistent.

6. Line 118-119, Figure 2: The HIV-1 reference sequences of CRF55_01B, CRF62_BC, CRF65_cpx, CRF68_01B and CRF69_01B is displayed the name of isolate, not the Genbank Accession Number. There are only one reference sequence for CRF55_01B, CRF62_BC, CRF65_cpx, CRF68_01B and CRF69_01B in the text, however, at least 3 reference sequences for each CRF in Figure2. Please clarify these questions.

7. Line 182, Line 254: “359” should be “358”.

8. Line 217: The font of the gene is recommended in italics, such as: pol.

9. Line 230, Line 247-250, Line 253: Place names and countries in English should be capitalized.

10. Is there a different HIV-1 drug resistance prevalence among recent infection and long-term infection? Please discuss the differences among them.

11. Figure 2: Bootstrap support value should be shown in the phylogenetic tree.

12. The format of references should be checked thoroughly.

13. Check English/grammar throughout.

6. PLOS authors have the option to publish the peer review history of their article (what does this mean?). If published, this will include your full peer review and any attached files.

Reviewer #1: No

Reviewer #2: No

---

## [Author Response · Author response to Decision Letter 0]

13 Oct 2020

Dear Editors and Reviewers:

Thank you for your letter and for the reviewers’comments concerning our manuscript entitled “The Genotype Distribution, Infection Stage and Drug Resistance Mutation Profile of Human Immunodeficiency Virus-1 among the Infected Blood Donors from Five Chinese Blood Centers, 2014-2017”. Those comments are all valuable and very helpful for revising and improving our paper, as well as the important guiding significance to our researches. We have studied comments carefully and have made correction which we hope meet with approval. Revised portion are marked in red in the paper. The main corrections in the paper and the responds to the reviewer’s comments are as flowing:

Responds to the reviewer’s comments:

Reviewer #1:

1. Response to comment: Materials and methods: The parameters for the phylogenetic analysis should be provided. The Genbank Accession Numbers of all obtained sequences should be provided

Response: We are very sorry for our negligence of the parameters for the phylogenetic analysis and the Genbank Accession Numbers of all reference sequences. We have provided the parameters for phylogenetic analysis in the complete sequence of the file . The Genbank Accession Numbers in line 111 -117 has been completed.

2. Response to comment: The authors mentioned that “Statistical analysis was performed using the chi-square test (or the Fisher's exact test) ” When the Fisher's exact test were used? In addition, “double-tailed test” should be “two-tailed test”..

Response: We are very sorry for our incorrect writing the chi-square test, A comparison of differences in drug resistance at the infection stage was added in line 251, using the Chi-square test.

3. Response to comment:Page 10, Line 184-189: The authors should carefully check the classification of DRM. for example, M184L belongs to NRTI, rather than NNRTI. “184M” should be written in a complete form.

Response: This problem was caused by our carelessness and has been rechecked and corrected between lines 179 and 184.

4. Response to comment: Page 12, Lines 239: Reference 26 did not support the statement that “The study also points out that these new CRFS are all from MSM and spread across regions through MSM groups”.

Response: After careful reading of literature 26, as the reviewer said, this literature does not support the statement that "this study also points out that these new CRFS are originated from gay men and spread across regions through gay men groups". This is our misunderstanding of the literature, we tried to find the literature, to support this view, I am sorry that could not be found, so we deleted this statement.

5. Response to comment: Page 12, Lines 246: How the DR prevalence rate ere (12.8%, 20/156) to be calculated?

Response: We are very sorry for our incorrect writing the rate of HIV-1 DR, HIV-1 DR rate should be 12.1% (12.1%, 43/356).

6. Response to comment: Page 12, Lines 249: The national HIV-1 drug resistance prevalence should be updated. Please see and cite the most recent Meta-analysis paper in EClinicalMedicine. 2020 Jan 5;18:100238.

Response: We have made correction according to the Reviewer’s comments on line 233.

7. Response to comment: Page 13, Lines 264: The statement that “the V179E may be transmitted” should be supported by phylogenetic analysis.

Response: This statement is our own subjective inference, we did not find reasonable evidence to prove it, so we deleted it.

8. Response to comment: The detailed demographic characteristics (age, gender, education, occupation and so on), subtypes, and DRMs of recruited patients should be provided.

Response: Thank you very much for the comments given by the reviewers. We focus on genotype this time. Not every positive sample can be sequenced successfully, so even if demogrphics is counted, it is not very meaningful.

9. Response to comment: S1 Table: Wrong classification of DRMs and incomplete drug sensitivity of patients.

Response: We are very sorry for the wrong classification of DRMs and incomplete drug sensitivity of patients. We have re-examined the classification of DRMs and added all the drug sensitivity of patients.(S1 Table).Subtypes and DRMs of recruited patients are provided in the table “China REDSIII HIV testing summary”.

10. Response to comment: Language can be further improved.

Response: We have further polished the article. As the reviewer said, the language really needs to be further strengthened, and we will make more efforts in this regard.

Reviewer #2:

Major comment

1. Response to comment: The HIV-1 subtype data shown in Table 1 and Figure 2 are not consistent. In Table 1 and Figure 2, the total number of Subtypes B, CRF69_01B and CRF62_BC were 15 and 17, respectively. Sequence CRF69_01B from Chongqing unmarked the color of node. The caption of CRF62_BC only including the reference sequences CRF62_BC, not the CRF62_BC sequence from Chongqing. In the subtree of CRF08_BC, CRF08_BC sequence from Urumqi unmarked the color of node. In the subtree of CRF55_01B, a total of 12 CRF55_01B sequences, however, only 11 CRF55_01B sequences showed in Table 1. In the subtree of CRF01_AE, one sequence belongs to CRF59_01B, however, no CRF59_01B appeared in the manuscript. In Table 1 and Figure 2, the number of CRF07_BC distribution in Mianyang and Guangxi were 16, 13 and 17, 11, respectively. There are two CRF_07 BC reference sequences not marked with node symbols. Please check the data carefully.

Response: Due to our mistakes, the data in Table 1 and figure 2 are inconsistent, and we have made changes. Through phylogenetic analysis, there are no genotypes of CRF69_01B and CRF62_BC in our study, and they have been corrected. Urumqi's CRF08_BC has been marked in figure2. Because we do not correspond to the data in table " China REDSIII HIV testing summary" and figure 2, the number of CRF55_01B is incorrect, which has been changed in table 1 and figure 2. although CRF59_01B was considered as a reconbinant form in this phygeny reconstruction, it appeared as a budding branch including one sample from Urumqi with a good bootstrap value among the 01_AE clade. As a result, we considered that one sample from Urumqi could be considered as 59_01B. And we clearly showed that in the pic 1. The number of CRF07_BC in Mianyang and Guangxi has been corrected. The two CRF_07BC reference sequences have been marked with node symbols.

2. Response to comment: In recent years, HIV-1 CRF55_01B has increased from the MSM population in China. In the present study, CRF55_01B was identified among blood donors in Chongqing (n=3), Urumqi (n=4), Luoyang (n=3) and Mianyang (n=1) (Table 1). I wonder there is the transmission link between the blood donors and MSMs, or link to CRF55_01B sequences from these regions and other region in China. So, I suggest authors try to perform a Bayesian analysis to infer the original time and transmission dynamics of the CRF55_01B of these blood donors, and elucidate their migration from these different regions in China or transmission between these two populations (blood donors and MSMs).

Response: Thanks a lot for this interesting question. We also want to investigate the original time when MSMs went to donate at the first time. However, as the amplyfied pol region was short, we suppose that Bayesian skyline plot analysis should uncertantly reveal the true population dynamics of this CRF. Luckly, we are recently colaborated with Shenzhen CDC where the CRF55_01B originated. So, we want to add more locations where CRF55_01B were sequnced, not only for the pol gene but rather the whole genome, that we can robustly investigate the original time and population dynamics of this specific interesting recombinant forms. Therefore, this interesting quesion could be better resolved.

Minor comment

1. Response to comment: Line 49: As the authors mentioned, “two PIs major DRMs (M46I)”, however, the PIs major DRMs shown in Table S1 is M46I/L. The authors should check this data.

Response: This problem was caused by our carelessness and has been rechecked and corrected between lines 179 and 184.

2. Response to comment: Line 50, Line 181: “thirty-four” should be “34”

Response: We are very sorry for our incorrect writing and we have corrected it.

3. Response to comment: Line52-53:CRF62_BC is identified from the 01B subtype? If so, the subtype information should be checked.

Response: CRF62_BC isn”t identified from the 01B subtype ,We have made correction on line 51-52 according to the Reviewer’s comments.

4. Response to comment: Line 66:“sub genotypes”should be “subtypes”or “genotypes”.

Response: We are very sorry for our incorrect writing and we have corrected it.

5. Response to comment: Line 84-89: The font of the paragraph needs to be reformatted, so that the paragraph format is consistent.

Response: We are very sorry for the negligence of this small problem, The font on lines 84-89 has been reformatted.

6. Response to comment: Line 118-119, Figure 2: The HIV-1 reference sequences of CRF55_01B, CRF62_BC, CRF65_cpx, CRF68_01B and CRF69_01B is displayed the name of isolate, not the Genbank Accession Number. There are only one reference sequence for CRF55_01B, CRF62_BC, CRF65_cpx, CRF68_01B and CRF69_01B in the text, however, at least 3 reference sequences for each CRF in Figure2. Please clarify these questions.

Response: We are very sorry for our negligence of the Genbank Accession Numbers of all reference sequences. We have fully provided the Genbank Accession Numbers in line 111 -117 .

7. Response to comment: Line 182, Line 254: “359” should be “358”.

Response: We verified the samples and finally included 356 samples, which have been changed in the manuscript.

8. Response to comment: Line 217: The font of the gene is recommended in italics, such as: pol.

Response: We are very sorry for this small mistake, and the manuscript has been checked and revised over and over again.

9. Response to comment: Line 230, Line 247-250, Line 253: Place names and countries in English should be capitalized.

Response: We have made correction according to the Reviewer’s comments on this question.

10. Response to comment: Is there a different HIV-1 drug resistance prevalence among recent infection and long-term infection? Please discuss the differences among them.

Response: It is really true as Reviewer suggested that we should discuss the differences among them. The results from 251th to 254th lines in the manuscript showed that P > 0.05, indicating that there was no difference in HIV-1 resistance between recent infection and long-term infection.

11. Response to comment: Figure 2: Bootstrap support value should be shown in the phylogenetic tree.

Response: In figure 2, we have marked out the Bootstrap support values that are greater than or equal to 60.

12. Response to comment: The format of references should be checked thoroughly.

Response: As Reviewer suggested that We have carefully examined the format of the references.

13. Response to comment: Check English/grammar throughout.

Response: I am very grateful to the reviewer for his opinion on us, and we have examined the problems of English grammar carefully .

We have tried our best to improve the manuscript and made some changes to it. These changes will not affect the content and framework of the paper. There are some minor problems that we did not list the changes, just marked in red in the revised paper. We sincerely thank the editors / reviewers for their enthusiastic work and hope that the revision of this article will be recognized. Thank you again for your comments and suggestions.

---

## [Decision Letter · Decision Letter 1]

5 Nov 2020

PONE-D-20-23288R1

The Genotype Distribution, Infection Stage and Drug Resistance Mutation Profile of Human Immunodeficiency Virus-1 among the Infected Blood Donors from Five Chinese Blood Centers, 2014-2017

PLOS ONE

Dear Dr. He,

Thank you for submitting your manuscript to PLOS ONE. After careful consideration, we feel that it has merit but does not fully meet PLOS ONE’s publication criteria as it currently stands. Therefore, we invite you to submit a revised version of the manuscript that addresses the points raised during the review process.

We look forward to receiving your revised manuscript.

Kind regards,

Chiyu Zhang, Ph.D.

Academic Editor

PLOS ONE

Reviewers' comments:

Reviewer's Responses to Questions

**Comments to the Author**

1. If the authors have adequately addressed your comments raised in a previous round of review and you feel that this manuscript is now acceptable for publication, you may indicate that here to bypass the “Comments to the Author” section, enter your conflict of interest statement in the “Confidential to Editor” section, and submit your "Accept" recommendation.

Reviewer #1: All comments have been addressed

Reviewer #2: All comments have been addressed

2. Is the manuscript technically sound, and do the data support the conclusions?

Reviewer #1: Yes

Reviewer #2: Yes

3. Has the statistical analysis been performed appropriately and rigorously? 

Reviewer #1: Yes

Reviewer #2: Yes

4. Have the authors made all data underlying the findings in their manuscript fully available?

Reviewer #1: Yes

Reviewer #2: Yes

5. Is the manuscript presented in an intelligible fashion and written in standard English?

Reviewer #1: Yes

Reviewer #2: Yes

6. Review Comments to the Author

Reviewer #1: (No Response)

Reviewer #2: The authors have addressed my comments and only a few minor changes to the manuscript are needed.

Minor comments:

1. This article needs to provide GenBank accession number of 356 obtained sequences.

2. Line 62: “CRF_08BC” should be “CRF08_BC”.

3. Line 113-114: “JP.AB845344_K231, JP.AB845345_K320; CRF62_BC: KC870035.1, KC870034.1, KC870037.1” should be “AB845344, AB845345; CRF62_BC: KC870035, KC870034, KC870037”.

4. Table 1: The sample size of CRF59_01B among five blood centers should be in regular font instead of bold.

5. Table 2: the total number of subtype B for recent infection should be “4”; the total number of subtype CRF01_AE for long-term infection should be “54”.

6. The sentence of line 201 should be further improved.

7. Line 218-220: The sentence (Finding substantial cases of these four CRFs) seems not accurate because the numbers of CRF55_01B, CRF59_01B and CRF68_01B were 15, 1, and 1, respectively. In addition, “four CRFs” should be “three CRFs”.

8. Line 228: “12.1% (12.1%, 43/356)” should be “12.1% (43/356)”.

9. Table 3: “250” should be “251”. “312” should be “313”. “355” should be “356”. The “P value” should be re-calculated.

10. The sentence of line 253-256 should be further improved.

11. Line 319: “A” should be “a”.

12. The format of references 8, 53, 54, 55 and 56 should be checked.

7. PLOS authors have the option to publish the peer review history of their article (what does this mean?). If published, this will include your full peer review and any attached files.

Reviewer #1: No

Reviewer #2: No

---

## [Author Response · Author response to Decision Letter 1]

21 Nov 2020

Dear Editors and Reviewers:

Thank you for your letter and for the reviewers’comments concerning our manuscript entitled “The Genotype Distribution, Infection Stage and Drug Resistance Mutation Profile of Human Immunodeficiency Virus-1 among the Infected Blood Donors from Five Chinese Blood Centers, 2014-2017”. Those comments are all valuable and very helpful for revising and improving our paper, as well as the important guiding significance to our researches. We have studied comments carefully and have made correction which we hope meet with approval. Revised portion are marked in red in the paper. The main corrections in the paper and the responds to the reviewer’s comments are as flowing:

Responds to the reviewer’s comments:

Reviewer #2:

1.Response to comment: This article needs to provide GenBank accession number of 356 obtained sequences.

Response: We are very sorry for our negligence of the Genbank Accession Numbers of 356 obtained sequences. We have provided GenBank accession number of 356 obtained sequences in line 129-131 of the manuscript.

2.Response to comment: Line 62: “CRF_08BC” should be “CRF08_BC”.

Response: We are very sorry for our incorrect writing the CRF08_BC. We have corrected it on line 62 of the manuscript.

3.Response to comment: Line 113-114: “JP.AB845344_K231, JP.AB845345_K320; CRF62_BC: KC870035.1, KC870034.1, KC870037.1” should be “AB845344, AB845345; CRF62_BC: KC870035, KC870034, KC870037”.

Response: This problem was caused by our carelessness and has been rechecked and corrected between lines 112 and 113.

4.Response to comment: Table 1: The sample size of CRF59_01B among five blood centers should be in regular font instead of bold.

Response: We are very sorry for our incorrect writing and we have corrected it.

5.Response to comment: Table 2: the total number of subtype B for recent infection should be “4”; the total number of subtype CRF01_AE for long-term infection should be “54”.

Response: We are very sorry for this small mistake, and the manuscript has been checked and revised over and over again.

6. Response to comment: The sentence of line 201 should be further improved.

Response: We have further polished this sentence. As the reviewer said, the language really needs to be further strengthened, and we will make more efforts in this regard.

7. Response to comment: Line 218-220: The sentence (Finding substantial cases of these four CRFs) seems not accurate because the numbers of CRF55_01B, CRF59_01B and CRF68_01B were 15, 1, and 1, respectively. In addition, “four CRFs” should be “three CRFs”.

Response: As the reviewer said, this sentence is indeed inaccurate, and we have revised it.

8. Response to comment: Line 228: “12.1% (12.1%, 43/356)” should be “12.1% (43/356)”.

Response: We have made correction according to the Reviewer’s comments on line 231.

9. Response to comment: Table 3: “250” should be “251”. “312” should be “313”. “355” should be “356”. The “P value” should be re-calculated.

Response: Thank you very much for the comment given by the reviewer. In the S1 Excel. China REDSIII HIV testing summary provided by us, the sample MY-12-003108 was not tested the infection stage, so this sample was excluded from the calculation.

10. Response to comment: The sentence of line 253-256 should be further improved.

Response:we are very grateful to the reviewer for his opinion on us, and we have examined the problems of English grammar carefully .

11. Response to comment: Line 319: “A” should be “a”.

 Response: This problem was caused by our carelessness and has been rechecked and corrected at line 322.

12. Response to comment: The format of references 8, 53, 54, 55 and 56 should be checked.

Response: We have made correction according to the Reviewer’s comments on this question.

We have tried our best to improve the manuscript and made some changes to it. These changes will not affect the content and framework of the paper.We sincerely thank the editors / reviewers for their enthusiastic work and hope that the revision of this article will be recognized. Thank you again for your comments and suggestions.

---

## [Editor Report · Decision Letter 2]

25 Nov 2020

The Genotype Distribution, Infection Stage and Drug Resistance Mutation Profile of Human Immunodeficiency Virus-1 among the Infected Blood Donors from Five Chinese Blood Centers, 2014-2017

PONE-D-20-23288R2

Dear Dr. He,

We’re pleased to inform you that your manuscript has been judged scientifically suitable for publication and will be formally accepted for publication once it meets all outstanding technical requirements.

Kind regards,

Chiyu Zhang, Ph.D.

Academic Editor

PLOS ONE
---

## [Editor Report · Acceptance letter]

11 Dec 2020

PONE-D-20-23288R2 

The Genotype Distribution, Infection Stage and Drug Resistance Mutation Profile of Human Immunodeficiency Virus-1 among the Infected Blood Donors from Five Chinese Blood Centers, 2014-2017 

Dear Dr. He:

I'm pleased to inform you that your manuscript has been deemed suitable for publication in PLOS ONE. Congratulations! Your manuscript is now with our production department. 

Kind regards, 

on behalf of

Dr. Chiyu Zhang 

Academic Editor

PLOS ONE